# Therapeutic Horizons: Gut Microbiome, Neuroinflammation, and Epigenetics in Neuropsychiatric Disorders

**DOI:** 10.3390/cells14131027

**Published:** 2025-07-04

**Authors:** Shabnam Nohesara, Hamid Mostafavi Abdolmaleky, Ahmad Pirani, Sam Thiagalingam

**Affiliations:** 1Department of Medicine (Biomedical Genetics), Boston University Chobanian & Avedisian School of Medicine, Boston, MA 02118, USA; snohesar@bu.edu (S.N.); sabdolma@bidmc.harvard.edu (H.M.A.); 2Department of Medicine, Beth Israel Deaconess Medical Center, Harvard Medical School, Boston, MA 02115, USA; 3Mental Health Research Center, Psychosocial Health Research Institute, Iran University of Medical Sciences, Tehran 14535, Iran; a.pirani85@gmail.com; 4Department of Pathology & Laboratory Medicine, Boston University Chobanian & Avedisian School of Medicine, Boston, MA 02118, USA

**Keywords:** gut microbiota, neuropsychiatric disorders, immune system, epigenetic, nutrition, inflammation

## Abstract

Neuroinflammation is a hallmark of many neuropsychiatric disorders (NPD), which are among the leading causes of disability worldwide. Emerging evidence highlights the significant role of the gut microbiota (GM)–immune system–brain axis in neuroinflammation and the pathogenesis of NPD, primarily through epigenetic mechanisms. Gut microbes and their metabolites influence immune cell activity and brain function, thereby contributing to neuroinflammation and the development and progression of NPD. The enteric nervous system, the autonomic nervous system, neuroendocrine signaling, and the immune system all participate in bidirectional communication between the gut and the brain. Importantly, the interaction of each of these systems with the GM influences epigenetic pathways. Here, we first explore the intricate relationship among intestinal microbes, microbial metabolites, and immune cell activity, with a focus on epigenetic mechanisms involved in NPD pathogenesis. Next, we provide background information on the association between inflammation and epigenetic aberrations in the context of NPD. Additionally, we review emerging therapeutic strategies—such as prebiotics, probiotics, methyl-rich diets, ketogenic diet, and medications—that may modulate the GM–immune system–brain axis via epigenetic regulation for the prevention or treatment of NPD. Finally, we discuss the challenges and future directions in investigating the critical role of this axis in mental health.

## 1. Introduction

Neuropsychiatric disorders (NPD) are brain diseases with a complex nature and great degree of genetic heterogeneity that influence millions of people around the world [1].

NPD are conditions that influence both the nervous system and mental health by altering a person’s thoughts, emotions, and behaviors [2]. NPD include neurodevelopmental disorders (e.g., autism spectrum disorder (ASD)), neurodegenerative diseases (e.g., Parkinson’s disease (PD), Alzheimer’s disease (AD)), and psychiatric disorders (e.g., bipolar disorder (BD), schizophrenia (SCZ)) that are commonly characterized by cognitive impairment, behavioral abnormalities, and disturbed emotional and mental functioning [3,4]. Neurodegenerative diseases such as AD and PD are characterized by the progressive loss of neuronal cells and disturbances in their function which in turn give rise to psychological disability and cognitive impairment [5]. However, mental illnesses are associated with dysfunctions in biological factors such as neuromodulators and neurotransmitters that primarily influence emotion, cognition, and behavior [6,7]. Neurodegenerative diseases typically develop over time, and the worsening of symptoms is associated with the disease progression, while mental illnesses are chronic or episodic, with different degrees of severity and duration.

The dysfunction of immune system has been known to be a key player in both the pathogenesis and the treatment outcomes of NPD [8]. The severity of clinical symptoms in NPD is linked to both peripheral and central inflammation [9]. Higher inflammatory responses in patients are connected to poorer response to antipsychotics [10]. Some events of the immune response such as chronic activation of macrophages and T cells, imbalance of T helper cells (Th1 and Th2), and elevated activation of microglia (the brain macrophages) during NPD, lead to impaired neurotransmitter release, derangements in synaptic plasticity and cortisol levels, which in turn may increase the burden of brain diseases [11]. The relationship between NPD and inflammation can be mediated by epigenetic alterations [12]. Accumulating evidence have indicated that aberrations in epigenetic mechanisms, such as DNA methylation, histone modifications, and non-coding RNAs (e.g., microRNAs (miRNAs) and long non-coding RNAs (lncRNA)), contribute to dysregulation of gene expression and determination of immune cell fate [13,14]. For example, the activity of histone deacetylases (HDACs) can regulate macrophage differentiation without lymphocyte stimuli and reduces macrophage plasticity, which further facilitates the pro-inflammatory phenotype [15]. Environmental factors such as lifestyle, diet, gut dysbiosis, and traumatic experiences have potential roles in inducing neuroinflammation and triggering epigenetic shifts, activation or suppression of pertinent genes associated with the onset and/or development of NPD [16,17].

One of the important determinants of health and disease in human is trillions of microorganisms such as bacteria, viruses, fungi, and other microorganisms that are inhabitant in different body organs specially the gastrointestinal (GI) tract [18,19]. Several studies demonstrated that gut bacteria-host interplays determine immune cell maturation, phenotype, and even migration and play a key role in neuroinflammation, the secretion of cytokines, chemokines, neurotransmitters, and microbial by-products [20]. For example, Lopizzo et al. found a specific inflammatory pattern in various brain regions, including dorsal and ventral hippocampus, that was linked to alterations in the gut microbiota (GM) composition [21]. Apart from the GM, altered levels of its metabolites are also connected to the immune-inflammatory response in the central nervous system (CNS) in NPD. For example, short-chain fatty acids (SCFAs) play a key role in regulating microglia maturation, and altered fecal SCFAs concentrations are connected to subclinical inflammation and impaired cognitive function in SCZ subjects [22,23]. Chen et al., also found a significant correlation between the abundance of pro-inflammatory bacteria, including *Clostridiales bacterium NK3B98* and *Ruminococcus* sp. *AM07-15* and altered fecal and plasma levels of SCFAs, including acetate, propionate, and butyrate in PD [24]. In AD, a reduction in the abundance of butyrate-producing bacteria like *Faecalibacterium* and raise in the abundance of lactate-producing genera contributed to immune disturbances and heightened neuroinflammation in the host [25].

Accumulating evidence linking the GM-induced inflammation or neuroinflammation to NPD through the GM–immune–brain axis has paved the way for developing therapies targeting this axis. This review provides an overview of the interplays between the GM, the immune system, and the brain via epigenetic aberrations in NPD. Moreover, various therapeutic interventions, such as prebiotics, probiotics, methyl-rich diets, and medications are highlighted for their role in modulating the GM–immune system–brain axis and alleviating these disorders.

## 2. Neuroinflammation in the Pathogenesis of Neuropsychiatric Disorders

One of the important etiological factors in NPD is neuroinflammation that may be caused by diverse pathogens or chronic inflammatory diseases [26]. During NPD pathogenesis, perturbations of the blood–brain barrier (BBB) facilitate the entry of peripheral immune cells and inflammatory molecules into the brain. In AD, inflammatory mediators change synaptic function and stimulate neurons and glial cells toward more Aβ production, which in turn lead to cognitive impairment [27]. In PD, dementia risk is connected to neuroinflammation. For example, Kouli et al. found that PD patients with higher dementia risk possess higher inflammation and/or tau accumulation prior to significant decline in cognitive performance [28]. In BD, higher amounts of IL-6 and IL-1β in the hippocampus are associated with more severe neuropsychiatric symptoms [29].

As immune cells and their inflammatory molecules are key players in the pathogenesis of NPD, there are significant changes in the expression of inflammatory genes. Evidence suggests that neuroinflammation in the pathogenesis of NPD is associated with epigenetic and transcriptional alterations in the peripheral immune system. For example, Ramakrishnan et al. found that in AD peripheral immune cells possessed more open chromatin and CD8 T cells exhibited a chromatin modification relevant to CXCR3 expression [30]. Their results also showed that in AD monocytes possessed APOE genotype-specific chromatin modifications, and genes relevant to sporadic AD exhibit chromatin-level changes [30]. Murphy et al. found elevated leukocyte expression of the NF-κB-activating receptors, TLR4 and TNFR2, and the NF-κB subunit, RelB, as well as reduced leukocyte expression of *IKKβ* and *NIK* mRNAs in SCZ subjects with higher levels of inflammation [31]. Additionally, the initiation of neuroinflammatory events in autistic BTBR mice is connected to elevation of alarmin S100A9 in the microglia [32]. A recent study indicated that induced pluripotent stem cells (iPSC)-derived astrocytes from patients with autism also exhibited overexpression of immune relevant genes, such as *TGFB1*, *TGFB2*, and *IL6* that was associated with DNA hypomethylation of *TGFB2*, *IL6*, and *TNFA* gene promoter regions [33]. Megagiannis et al. reported that ASD-associated CHD8 in astrocytes play an important role in the injury-induced reactive gliosis by remodeling chromatin accessibility and alerting gene expression; therefore, targeting CHD8 may represent a promising strategy to alleviate neuroinflammation [34]. Moreover, neuroinflammation-induced depression-like behaviors in mice has been linked to reduced hydroxymethylation of *BDNF* gene and elevated expression of Iba-1, a marker for microglia activation, in the hippocampus region [35]. Additional studies on altered immune cell composition or inflammatory molecules in specific NPD, along with their consequences, are summarized in Table 1.

## 3. The Gut Microbiome May Orchestrate the Function and Behavior of Immune System

Several lines of evidence indicates that the GM is a master regulator in modulating CNS inflammation, and disruption in GM composition in NPD is linked to substantial alterations in gut anti-inflammatory and pro-inflammatory bacterial genera [46]. As an interesting example, a subset of astrocytes referred to as “gut-licensed IFNγ+ NK cells drive LAMP1+ TRAIL+ anti-inflammatory astrocytes” in mice can prevent CNS inflammation by the induction of T cell apoptosis mediated by TRAIL–DR5 signaling [47]. While, GM modulates the production of IFNγ (an anti-inflammatory cytokine) by meningeal NK cells, IFNγ induces TRAIL protein expression (encoded by *TNFSF10* gene) in astrocytes. However in inflammation, T cells and microglia produce some molecules, which suppress astrocytic expression of TRAIL protein [47]. In other studies, it appears that lactic acid-producing bacteria may play a role in regulation of the function of other immune cells, specially macrophages, since lactate has shown capability for supporting a metabolic–epigenetic link in macrophage polarization [48]. Ling et al. observed that structural dysbiosis in SCZ patients is relevant to the compositional changes in several genera involved in producing immunomodulatory metabolites, which further elevated the amounts of pro-inflammatory cytokines, such as IL-1β and reduced levels of anti-inflammatory cytokines, like IFN-γ [49]. Huang et al. examined the compositional changes in GM and its relation to inflammation in depressed BD patients and found reduced levels of some bacteria (*Butyricicoccus*, *Lachnospiraceae incertae sedis*, and *Dorea*) involved in triggering the secretion of anti-inflammatory cytokines such as IL-4, IL-10, and IL-11 [50]. Lower levels of the anti-inflammatory butyrate-producing bacteria was also reported in untreated BD patients versus healthy controls [51]. Similarly, changes in butyrate-producing genera have been found in SCZ patients [52]. The authors mentioned that patients with SCZ exhibited the depletion of anti-inflammatory butyrate-producing genera, and elevated abundance of some opportunistic bacteria genera [52]. PD is also associated with decreased abundance of anti-inflammatory butyrate-producing bacteria, such as *Roseburia*, *Faecalibacterium*, *Blautia*, and *Lachnospira* and increased abundance of pro-inflammatory genera such as *Streptococcus* [53]. Figure 1 depicts how disruption in GM composition due to gut dysbiosis generally contribute to increased neuroinflammation, which in turn promotes the development of NPD.

More studies addressing the relationship between pro- and anti-inflammatory gut bacteria and different NPD are presented in Table 2.

In addition to microbial communities in the GI, metabolites produced by commensal gut flora may mitigate pathogenic activities of microglia and astrocytes by controlling microglia activation and regulating astrocytic transcriptional program via a mechanism mediated by the aryl hydrocarbon receptor [68]. GM-derived metabolites may also play a powerful role in the modulation of the function of immune cells by orchestrating epigenetic modifications. For example, the anti-inflammatory effects of butyrate is associated with the modulation of Mψ (M1/M2) polarization by decreasing the activity of HDAC1 and negative regulation of miR-155 [69]. Moreover, the exogenous acetate produced by some bacteria can contribute to rescuing defective M2 polarization and histone acetylation following mitochondrial pyruvate carrier 1 (MPC1) inhibition or in adenosine triphosphate–citrate lyase (ACLY) deficiency [48]. In another study, both patients with SCZ and major depressive disorder (MDD) exhibited reduced levels of acetic acid, isobutyric acid, and propionic acid, which were associated with higher Positive and Negative Syndrome Scale (PANSS) and Hamilton Depression Rating Scale (HAMD) scores, increased levels of pro-inflammatory cytokines, and reduced levels of anti-inflammatory cytokines (IL-37 and IL-10) [70]

## 4. Epigenetic Alterations Linked to Immune System and Immune-Relevant Genes in Neuropsychiatric Disorders

### 4.1. DNA Methylation Patterns

Patients with NPD exhibit noticeable changes in DNA methylation patterns that may affect immune cell composition. For example, Liu et al. reported that the most evident biological functions of differentially methylated CpGs in SCZ subjects are related to inflammatory response of *CD224*, *LAX1*, *TXK*, *PRF1*, *CD7*, *MPG*, and *MPO* genes that are responsible for activations of T cells, B cells, and natural killer cells [71].

Shindo et al. reported that subjects with untreated MDD exhibited decreased levels of natural killer cells along with epigenetic aging acceleration in HannumAge and GrimAge (two epigenetic clocks—biomarkers representing 71 and 1030 CpG sites, respectively) in blood samples [72]. In a study by Luo et al. a subtype of SCZ consisting almost 40% of patients with higher symptom severity, worst cognitive function, and gray matter thickening exhibited extensive DNA methylation changes among genes linked to immune cell activity, an increase in the proportion of neutrophils, and a decrease in the proportion of lymphocytes [73]. Additionally, they reported that a subgroup of never-treated first-episode SCZ patients (47.5%) had also widespread methylation changes in genes of immune cell activity and increased proportion of neutrophils in blood samples [74]. Table 3 summarizes more studies about associations between altered methylation patterns in immune-relevant genes and developing NPD in humans.

### 4.2. Histone Modifications

Histone modifications can also contribute to controlling gene expression and regulating inflammatory responses during NPD pathogeneses. The activation of microglia (i.e., inflammatory microglia) confers a sequential chain of events, including hyperexpression of *HDAC2* in neurons, reducing amounts of histone acetylation, and inhibiting the transcription of *BDNF* and *c-Fos*, which results in memory impairment [88]. The activated microglia can also arrest Nrf2-inducible anti-oxidant defense in astrocytes, which is connected to reduction in histone acetylation [89]. Additionally, the hypothalamic and hippocampal transcription of pro-inflammatory targets by lipopolysaccharide (LPS) in neurons and reactive microglia is mediated by increasing the level of H3S10phK14ac (H3 phosphoserine10 and lysine 14 acetylation) in rat brain [90]. Rodriguez-Zas et al. also reported that inflammation-associated depression-like behaviors in mice are linked to derangements in microglia histone acetylation [91]. Additional examples related to associations between altered histone modifications in immune-relevant genes linked to NPD are provided in Table 4.

### 4.3. MicroRNAs (miRNAs)

miRNAs, the endogenous and small non-coding RNA molecules (21–25 nucleotides in length), are among other main regulators of gene expression at post-transcriptional level through incomplete pairing with the 3′-untranslated regions (3′-UTRs) of target mRNAs [92]. Numerous inflammatory miRNAs like miR-155 and let-7 play key roles in stimulating neuroinflammation by binding to toll-like receptors (TLRs), activation of microglia and astrocytes, and release of inflammatory mediators [93,94,95]. For example, miR-145-5p is a negative regulator of astrocyte proliferation, and its down-regulation increases smad3 activity and subsequently astrocyte proliferation [96]. There is also a report indicating that elevated levels of miR-223 in the orbitofrontal cortex of SCZ and BD patients are positively and negatively associated with inflammatory and GABAergic gene expression, respectively [97]. Furthermore, Kaurani et al. reported that reduction in the miR-99b-5p level in mice could cause both SCZ-like phenotypes and inflammatory processes, which are relevant to synaptic pruning by microglia [98]. Table 4 summarizes additional studies on the associations between altered miRNAs in immune-relevant genes involved in the pathogenesis of NPD.

**Table 4 cells-14-01027-t004:** Altered miRNAs and histone modifications of immune-relevant genes in specific neuropsychiatric conditions in humans or animal studies.

Neuropsychiatric Disorders/Study Subjects	Epigenetic Study/Samples	Immune-Relevant Genes	Key Findings	Ref.
SCZ/human	Histone acetylation/peripheral blood mononuclear cells (PBMCs)	IL-6 and IFN-γ	Histone H4 hypoacetylation in PBMCs decreases IL-6 and IFN-γ following 90 days exercise in SCZ vs. day 0.	[99]
MDD/human	Histone acetylation/dentate gyrus	ISG15, IFI44L, IFI6, NR4A1/Nur-77, GABBR1, CCL2/MCP-1, KANSL1	Differential expression of 30 genes involved in histone acetylation and inflammation (e.g., CCL2 and KANSL1 in MDD)	[100]
MDD/human	Histone methylation/peripheral blood cells (PBCs)	TNFAIP3, TLR4, TNIP2, miR-146a, miR-155	Lower histone 3 lysine 4 tri-methylation (H3K4me3) levels at the promoters of TNFAIP3, TLR4, TNIP2, miR-146a, and miR-155 in MDD	[101]
Alzheimer’s disease (AD)/APP/PS1 mice	Histone acetylation/Entorhinal Cortex	CREB, IL-1β, and TNF-α and NF-kB	Reduced H3K9K14 acetylation, increasing Aβ deposition, microglia and astrocytes activation, and inflammatory factors via the CREB/BDNF/NF-kB pathway	[102]
Postpartum psychosis/human	miR-146a and miR-212/monocytes	ADAM17, EGR3, IRAK2, PTGS2, CXCL2, and PTGS2	Reduced miR-146a expression in monocytes diminishes natural T regulator cells; reduced expression of miR-212 elevated Adrenomedulin, reduced IL-6, and increased Th2 cells	[103]
SCZ/human	miR-337-3p, miR-127-5p, miR-206, miR-1185-1-3p/human iPSC-derived astrocytes from SCZ patients	IL-1β, LAMTOR4, IL23R, ERBB3, ERBB2, and IRAK1	Lower expression of miR-337-3p, miR-127-5p, miR-206, and miR-1185-1-3p in SCZ astrocytes	[104]
SCZ/human	hsa-miR-16-5p, hsa-miR-186-5p, hsa-miR-19a-3p, and hsa-miR-19b-3p/blood	IL-1β, IL-6, and TNFα	Higher PANSS scores is linked to down-regulation of four miRNAs that negatively regulate pro-inflammatory cytokines	[105]
Bipolar Disorder (BD)/human	hsa-miR-34a-5p, hsa-miR-152-3p hsa-miR-574-3p, hsa-miR-3128 and hsa-miR-3201/Lymphoblastoid cell lines	NF-κB, STAT3, and TNF	46 up-regulated and 31 down-regulated miRNAs with immune-related functions in responders vs. non-responders to Lithium in BD	[106]
MDD/human	let-7e, miR-21-5p miR-145, miR-223, miR-146a, and miR-155/PBMCs and monocytes	TLR4	Lower levels of let-7e, miR-146a, and miR-155 in PBMCs and miR-146a and miR-155 in monocytes in MDD vs. controls	[107]
MDD/human	miRNA-144-5p/plasma	CXCL6, STAMPB, CXCL1, CXCL5, IL-7, MCP-4, MCP-2, MCP_1, MMP-1, IL-8, and IL-18	An inverse correlation between miR-144-5p and some inflammatory proteins	[108]
MDD/human	miR-342, miR-146a, and miR-155/PBMCs and plasma	TNF-α, IL-6, and CCL2 (plasma)	Positive correlation between miR-342 expression and TNF-α level	[109]
Parkinson’s disease (PD)/mice	miR-335/serum samples	LRRK2	Mitigating neuroinflammation by miR-335 via targeting LRRK2	[110]
Autism/mice	miRNA profiling/prefrontal cortex (PFC)	NF-κB, IRAK1, and TLR7	Neuroinflammation is linked to miR-146a, let- 7b, and miR-592	[111]
AD/mice	Several miRNAs/PFC and the hippocampus	Cst7 and Gfap	Neuroinflammation is linked to miR-124-3p, miR-125b-5p, miR-21-5p, miR-146a-5p, and miR-155-5p	[112]

## 5. Therapeutic Approach in Neuroinflammatory Diseases

Considering interconnected links between GM, neuroinflammation, and epigenetic alterations in NPD, there are opportunities to target these key interlinked players for the treatment of these diseases. Potential remedies may include prebiotics, probiotics (i.e., nutrition that tailor GM), postbiotics (products of GM) to alleviate inflammation and epigenetic alterations, in addition to psychiatric drugs, which in part influence GM, neuroinflammation and the epigenome.

### 5.1. Probiotics

Probiotics are microorganisms capable of exerting health benefits when are utilized or consumed in sufficient amounts [113]. Anti-inflammatory effect of probiotics may be associated with enhancing Na^+^-dependent absorption of luminal butyrate, reshaping GM composition, and altering the expression of genes involved in immune reactions in the prefrontal cortex [114,115]. In a study published in the early 2010 s, the beneficial effects of SCFA-producing bacteria *Lactobacillus helveticus R0052* and *Bifidobacterium longum R0175* on improving signs of post-myocardial infarction depression in rats were linked to a significant reduction in circulating IL-1β levels and the restoration of intestinal barrier integrity [116]. *Lactobacillus plantarum* MTCC 9510 supplementation could also mitigate depressive-like behaviors in mice exposed to chronic unpredictable mild stress (CUMS) by reducing serum TNF-α and LPS levels, elevating the abundance of *Lactobacillus* sp. decreasing the abundance of *Enterobacteriaceae*, and increasing the production of acetate and butyrate [117].

In another study, researchers served *Clostridium butyricum*, a butyrate-producing, spore-forming anaerobic bacterium, as a probiotic to examine its antidepressant-like effect in a mouse model of inflammatory depression [66]. Their results indicated that this probiotic could improve depressive-like behaviors by reducing inflammatory factors and normalizing the GM composition [66]. Parra et al. reported that oral administration of a combination of two probiotics, including *Lactobacillus rhamnosus GG* and *Bifidobacterium animalis* ssp. *lactis Bb12* once daily for 15 days in an inflammatory PD model in rats could improve symptoms by decreasing microglial activation [118]. Valvaikar et al. examined protective effect of probiotic strain, *B. breve Bif11*, in the 1-methyl-4-phenyl-1,2,3,6-tetrahydropyridine (MPTP)-induced PD model in rats and their findings showed improvements in the cognitive and motor abnormalities, reductions in pro-inflammatory mediators, oxidative and nitrosative stress in the mid brain of MPTP-lesioned rats by elevating the concentration of epigenetic metabolites, including butyric acid, propionic acid, and acetic acid [119]. Commercial probiotic VSL#3^®^, including eight bacterial strains (including, different *Lactobacillus* and *Bifidobacterium* as well as *Streptococcus thermophilus BT01*) could ameliorate autistic-like behaviors in a prenatal valproic acid-induced rodent model of autism by elevating serum concentration of the IL-10 and reducing IL-6 [120]. In a clinical study by Reininghaus et al., a four-week probiotic plus biotin supplementation could improve symptoms in MDD patients by increasing the abundance of butyrate-producing bacteria, including *Ruminococcus gauvreauii* and *Coprococcus 3* and by reducing inflammation [121]. In another clinical study a 12-week intervention with a multi-strain probiotic (containing *Lactobacillus plantarum PL-02*, *Bifidobacterium longum subsp. infantis BLI-02*, *B. animalis subsp. lactis CP-9*, *B. bifidum VDD088*, and *B. breve Bv-889*) resulted in a 36% increase in BDNF, and anti-oxidant SOD levels but decrease in IL-1β levels in AD [122]. A different recent clinical study indicated a 12-week intervention with two single-strain probiotics (*Lacticaseibacillus rhamnosus HA-114* or *Bifidobacterium longum R0175*) in subjects with mild and moderate AD could improve the serum concentration of GSH, IL-10, and reduce the serum concentration of 8-OHdG, MDA, TNF-α, and IL-6, possibly through producing epigenetic metabolites like butyric acid and propionic acid [123].

### 5.2. Prebiotics/Postbiotics

Prebiotics are ingredients in food that enhance growth or activity of commensal bacteria [124]. Prebiotics have demonstrated ability for exerting protective effect by reducing mucosal inflammation, enhancing the colonic production of SCFAs involved in mitigating epigenetic aberrations, and regulating the GM via SCFA-induced inhibition of M1 and activation of M2 macrophages [125,126,127,128]. In another study Guo et al. reported that the anti-inflammatory effect of inulin (a soluble dietary fiber) on mice with SCZ phenotype is associated with enhancing intestinal integrity via elevating the expression of tight junction proteins, diminishing inflammatory cytokines, increasing BDNF levels, and raising the abundance of *Lactobacillus* and *Bifidobacterium.* These effects are due to the production of anti-inflammatory metabolites, such as butyrate, which modulate the epigenome [129]. Inulin has also been shown to improve behavioral abnormalities in chronically stressed mice by enhancing the production of SCFAs, mitigating CUMS-induced increases in BBB permeability, preventing LPS penetration into the brain, and regulating TLR4/MyD88/NF-κB signaling pathway to attenuate inflammatory responses [130]. Additionally, a 10-week oral natural prebiotics supplementation, including topinambur powder and chicory root inulin could improve anxiety-depressive behaviors, cognitive disorders and dysbiosis in mice exposed to CUMS by elevating the number of bacteria, such as *Ruminococcaceae UCG-014*, *Odoribacter*, *Roseburia*, *Anaerotruncus*, *Intestinimonas*, *Lachnospiraceae ASF 356*, and *Clostridielesvadin BB60* involved in the production of metabolites with epigenetic effects [131]. Dietary inulin could also improve depressive and anxiety-like behaviors in constipation induced depression in mice by suppressing neuroinflammation, hampering synaptic ultrastructure damage, reducing the abundance of *Muribacalum* and *Melaminabacteria*, and elevating the concentration of different SCFAs like acetate, propionate, and butyrate in the feces [132].

In a recent human study Buchanan et al. reported that a 10-day oligofructose-enriched inulin treatment in SCZ patients could increase plasma butyrate level, which was associated with reduced inflammation [133]. Treatment with Zhi-Zi-Chi decoctions (ZZCD), consisting of Gardeniae Fructus and Semen sojae praeparatum was shown to improve depressive-like behaviors in rats exposed to a depression model of CUMS by increasing the abundance of anti-inflammatory bacteria, diminishing inflammatory and tryptophan-metabolizing bacteria, reducing pro-inflammatory cytokines, and restoring butyrate in the cecal content [134]. Another animal study examined the effect of a prebiotic, including Fructooligosaccharides (FOS) and Galactooligosaccharides (GOS) on inflammatory response and depressive and anxiety-like behaviors in a mouse model fed a high-fat diet (HFD) [135]. The results showed that treatment with FOS and GOS could improve depression and anxiety-like behavior by elevating the levels of acetate-producing bacteria (*B. acidifaciens* and *B. dorei*), increasing acetate and GPR43 concentrations in the brain, and subsequently ameliorating chronic peripheral and central inflammation [135]. Shu et al. reported that Xiaoyaosan polysaccharide alleviated depression-like behaviors by modulating the diversity of intestinal butyrate-producing bacteria diversity, including *Roseburia* sp. and *Eubacterium* sp. [136]. Another study also showed that antidepressant-like effects of Xiaoyaosan polysaccharides are linked to enriching the diversity and elevating the abundance of the butyrate-producing bacteria *Roseburia* sp. and *Eubacterium* sp. [137]. Additionally, raspberry ketone has been shown to reduce gut barrier dysfunction and hamper LPS-induced depression-like behaviors in mice by suppressing TLR-4/NF-κB signaling pathway involving the gut–brain axis and increasing the production of SCFAs [138]. In a valproic acid-induced mouse model of autism, a diet composed of prebiotic fibers, namely, GOS/FOS, could modulate peripheral immune homeostasis, attenuate neuroinflammation in the cerebellum, and improve deficits in social behavior and cognition by improving intestinal integrity, reshaping GM, and restoring the cecal concentrations of SCFA like propionic acid [139]. Sarti et al. also found that the symbiotic treatment with (a multi-extract of fibers and plant complexes, containing inulin/fruit-oligosaccharides) and probiotics (a 50%–50% mixture of *Lactobacillus rhamnosus* and *Lactobacillus paracasei*) could attenuate pathological features of AD, including the neuronal degeneration, Aβ deposition and neuritic plaques, and the shrinkage in the cortex of APP/PS1 mice by reducing *GFAP* expression and microglia activation [140].

In a clinical study conducted by Bedarf et al., a 4-week diet rich in dietary fiber, and the intake of Lactulose prebiotic, could improve PD-dependent gastrointestinal symptoms by elevating *Bifidobacteria* spp. and restoring epigenetic metabolites such as butyrate and propionate [141].

SCFAs are considered a class of postbiotics. These metabolic byproducts are produced by gut bacteria during the fermentation of dietary fibers, and act as master regulators in host health [142,143]. In the brain, SCFAs can exert anti-inflammatory effects via two major cellular mechanism [144]. The first mechanism involves SCFAs binding to G protein-coupled receptors (GPRs), increasing their activity, and thereby contributing to the maintenance of intestinal immune hemostasis. The second mechanism is the inhibition of HDAC activity, which leads to the suppression of inflammatory cytokines expression in the CNS [145]. Butyrate, a major SCFA produced by the GM through the fermentation of insoluble fibers has been shown to rectify the imbalance in microglia polarization (M1/M2), enhance the expression of intestinal tight junction protein occluding, maintain epithelium barrier integrity, suppress oxidative stress, and diminish LPS-induced inflammation in primary microglia, and neuronal cells in the hippocampus region, and neuronal cells co-cultured with microglial or astrocytes in rats [146,147,148,149,150]. Moreover, butyrate may exert its anti-inflammatory effect by promoting a balance between anti-inflammatory Treg and pro-inflammatory T helper 17 (Th17) cells [151]. As butyrate is involved in facilitating extrathymic generation of Treg cells [151], its anti-inflammatory effects are also mediated by activation of G protein receptor 109a, inhibition of TLR4 signaling and suppression of NF-κB activation [152,153].

In an animal model of acute mania induced by d-amphetamine (d-AMPH), it has been shown that sodium butyrate (SB) could inhibit d-AMPH-related hyperactivity and restore the activity of mitochondrial respiratory-chain complexes as well [154]. Qiu et al. established LPS-induced depression model in mice to examine protective effects of SB. Their findings showed that pretreatment at the dose of 300 mg/kg for 10 days could improve LPS-induced depressive-like behaviors and prevent LPS-induced elevation in pro-inflammatory cytokines, such as IL-1β, IL-6, and TNF-α [155]. In a most recent animal study Wang et al., demonstrated that butyrate could improve depression-like behaviors and mitigated ferroptosis of prefrontal cortex (PFC) neurons induced by acute stress in mice, through modulation of the gut–brain axis and reduction in systemic inflammatory responses [156]. These mechanisms collectively suggest that butyrate supplementation may serve as a promising candidate for alleviating psychological symptoms [157].

Propionate is another SCFA found in both the gut and the bloodstream. Its protective effects in pathological conditions are attributed to the improvement of intestinal barrier function, attenuation of inflammation and oxidative stress, primarily through the modulation of inflammation-related signaling pathways [158,159]. Preoperative supplementation with propionate could enhance intestinal function recovery in rats and ameliorate neuroinflammation and hippocampus-related memory impairment, due to propionate regulatory influence of on Th17 cell-mediated immune responses [160]. Propionate can also enhance neurocognitive function by reducing Th17 cells infiltration into the CNS, decreasing *IL-17A* expression, and suppressing microglial activation through an IL-17A/IL-17RA-dependent mechanism [160]. Propionate may contribute to boosting de novo Treg-cell generation in the periphery by inhibiting HDAC activity as well [151]. Intravenous administration of low-dose propionate (2 mg/kg body weight/day) was shown to exert antidepressant effects in rats exposed to CUMS [161]. Moreover, short term intrarectal administration of sodium propionate exhibited antidepressant-like effects in CUMS rats [162]. In a recent study, protective effects of oral sodium propionate in CUMS-exposed rats were attributed to reshaping GM and region-specific epigenetic modification of histone 3 [163].

### 5.3. Methyl Rich Diets and Inflammatory Responses

Some key products of the GM, such as folate and vitamin B12 may exert protective effects against neuroinflammatory disorders by modulating the GM–immune–brain axis, via DNA methylation alterations [164]. Deficiency of these essential products due to gut dysbiosis may contribute to the development of NPD by triggering inflammation. For example, folate deficiency mediates neuroinflammation by inducing NF-ĸB activation [165]. Therefore, their supplementation may be considered a promising strategy for the treatment of immunometabolic form of NPD [166]. In this context, Menegas et al. reported that folic acid, when used as a therapeutic adjunct to lithium, alleviate the manic-like behaviors, oxidative stress, and pro-inflammatory cytokines in a rat model of mania induced by methamphetamine (m-AMPH) [167]. Khosravi et al. found that a healthy dietary pattern decreases the risk of depression by increasing serum levels of the folate and vitamin B12, whereas an unhealthy dietary pattern increases the risk of depression by reducing their serum levels [168]. Jia et al. also reported that neuroprotective effect of folic acid in MPTP-induced PD in mice was linked to suppressing neuroinflammation by inhibiting NLRP3 inflammasome and improving mitochondrial integrity via the p53-PGC-1α pathway [169].

In a clinical study conducted by Chen et al., folic acid supplementation could alleviate AD symptoms by decreasing mRNA levels of pro-inflammatory cytokines such as TNFα in leukocytes [170]. Additionally, Chen et al. reported that folic acid and vitamin B12 supplementation could hamper the progression of cognitive decline in AD patients by decreasing serum concentrations of TNFα [164].

Choline, a methyl donor and precursor of the neurotransmitter acetylcholine, has also been considered a promising intervention for regulating the ratio of pro-to-anti-inflammatory cytokines in the hippocampus and modulating long-term inflammatory tone [171]. Dietary choline intake may be considered a promising candidate to alleviate hallmarks of AD, particularly inflammation [172]. Mechanistically, Egilmez et al. reported that ameliorative influence of choline chloride in the treatment of social behavior in a rat model of LPS-induced autism is connected to reduced levels of TNF-α, IL-2, and IL-17 [173].

In U.S. adults, dietary choline intake is negatively correlated to the risk of depressive symptoms [174]. Furthermore, betaine, one of the choline oxides in the body, may exert an anti-inflammatory effects by suppressing the NF-ĸB signaling pathway, a master regulator of genes involved in the inflammatory response associated with depression [175]. Another mechanism is that betaine helps reduce homocysteine levels by converting it to methionine, and hence exerting antidepressant effects, as elevated homocysteine is linked to an increased risk of depression in humans [176,177]. In addition to anti-inflammatory effects, protective role of betaine is associated with overexpression of tight junction proteins (occludin and ZO-1), promoting the intestinal barrier integrity, and modulation of the GM [178]. Liu et al. also found that betaine alleviates depression and anxiety-like behaviors in mice with dextran sulfate sodium-induced colitis by reducing DNA damage, improving mitochondrial impairment, and suppressing the cGAS-STING signaling pathway involved in the inflammatory responses [179]. Hui et al. reported that betaine could rescue METH-induced depressive-like behavior and cognitive dysfunction by reducing neuroinflammation through inhibition of NLRP3 inflammasome in hippocampal region of mice [180]. Moreover, betaine has demonstrated ability for suppressing AβO-induced neuroinflammation in microglia by mitigating the activation of the NLRP3 inflammasome and NF-κB [181].

### 5.4. Modified Mediterranean Diet and Ketogenic Diet

Certain specific diets have demonstrated capacity for improving NPD through inhibition of neuroinflammation or modification of the GM structure and composition, thereby affecting epigenetic mechanisms [182,183,184]. Joseph et al. proposed that protective effects of a modified Mediterranean diet promoting the production of SCFAs (e.g., acetate, butyrate, and propionate) by the GM are associated with improvements in immune and metabolic dysfunction in SCZ [185]. The ketogenic diet (KD) is another effective approach to improving certain NPD, such as epilepsy-associated depression, autism bipolar disorder, and SCZ [186]. The KD is a high-fat, low-carbohydrate, and moderate-protein diet that shifts the body’s metabolism toward using fat as its primary energy source instead of carbohydrates [187]. This metabolic state, called ketosis, leads to the production of ketone bodies (such as beta-hydroxybutyrate, an epigenetic modifier) that serve as an alternative fuel, particularly for the brain. In a recent mechanistic study Liang et al. demonstrated that KD could improve CUMS-induced depression-like behaviors in mice by suppressing glial activation markers (Iba-1 and GFAP), hampering the expression of inflammatory cytokines, including IL-1β, TNF-α, and COX-2, inhibiting the TLR4/MyD88/NF-κB signaling pathway, and promoting the BDNF/TrkB/CREB and Wnt/β-catenin signaling pathways [188]. In an AD mouse model, Di Lucente et al. reported that KD could rescue Long-Term-Potentiation (LTP), increase synaptic plasticity, and decrease markers of microgliosis by elevating the concentration of beta-hydroxy-butyrate [189]. Zhu et al. found that the dietary intervention with KD in a LPS-induced rat PD model could reverse the disease progression by preventing certain events, including overexpression of pro-inflammatory mediators (TNF-α, IL-1β, and IL-6), the loss of dopaminergic neurons, and diminishing mGluR5+ microglia cells, elevating TSPO+ microglia cells, decreasing H3K9 acetylation in the mGluR5 promoter region and mGluR5 mRNA expression, and reducing the phosphorylation levels of Akt/GSK-3β/CREB pathway [190].

In a clinical study Allan et al. demonstrated that a modified KD could exert protective effects in ASD children by altering the GM, elevating the expression of butyrate kinase in the gut, reducing the amount of pro-inflammatory cytokines (IL-12p70 and IL-1β), and changing the levels of BDNF-related miRNAs in the plasma [191]. In another clinical study conducted by Schweickart et al., the Mediterranean KD was capable of alleviating risk factors of AD by decreasing the inflammatory marker GlycA, suggesting a reduction in systemic inflammation owing to the effects of this type diet [192].

### 5.5. Immunomodulatory Effects of Different Drugs in the Treatment of Neuropsychiatric Disorders by Targeting GM via Epigenetic Mechanisms

Accumulating evidence has demonstrated that the beneficial effects of antipsychotics, antidepressants, and antibiotics in the cure of NPD are associated with their ability to down-regulate neuroinflammation by altering GM composition and increasing production of metabolites with anti-inflammatory and epigenetic effects. Figure 2 shows how different types of medications, including antipsychotics, antidepressants, and antibiotics contribute to the treatment of NPD by altering pro- and anti-inflammatory gut bacteria and changing the production of epigenetic metabolites with anti-inflammatory effects.

#### 5.5.1. Psychiatric Medications That Influence GM, Inflammation and the Epigenome

Besides their effects on neurotransmission, psychiatric drugs also exhibit anti-inflammatory properties and modulate the GM. For instance, patients with BD and treated with quetiapine (300 mg/d) for four weeks exhibited a rebound in the abundance of *Bifidobacteria* and *Eubacterium rectale* bacteria, which are involved in reducing inflammation by producing butyrate and other SCFAs [193]. In another study Li et al., found that after 24-week risperidone treatment, patients with SCZ exhibited a remarkable increase in serum levels of butyric acid, accompanied by improvements in the PANSS positive symptoms subscale scores [194]. A four-week treatment with amisulpride in SCZ patients was also associated with an increased abundance of SCFA-producing bacteria (such as *Dorea* and *Butyricicoccus*), and a decreased abundance of pathogenic bacteria (*Actinomyces* and *Porphyromonas*) and a reduction in IL-6 levels [195]. Another recent study demonstrated that treatment with amisulpride exerted anti-inflammatory effects in SCZ patients by elevating serum levels of acetate, increasing the expression of several genes, including *HDAC1*, *GPR109a*, *GPR43*, *TLR2*, soluble CD14 (*sCD14*), and N-methyl-d-aspartate receptor (*NMDAR*), but reducing the expression *TLR4* and pregnane X receptor (*PXR*) [196].

Mood stabilizers, a class of medications used for treating mood swings in BD patients, can also exert their protective effects by reshaping the GM and modulation of immune system function via epigenetic mechanisms. For example, lithium carbonate treatment could mitigate colon inflammation by modulating gut microbial diversity, increasing the abundance of SCFA-producing bacteria, such as *Akkermansia muciniphila*, and enhancing anti-inflammatory Treg cell activity in colonic lamina propria (LP) through a GPR43-related mechanism [197]. Furthermore, a more recent study showed that the recommended dose of lithium increases the abundance of SCFA-producing bacteria, including *Lactobacillus*, *Ruminococcaceae*, *Enterorhabdus*, *Muribaculaceae*, and *Coprococcus* [198].

#### 5.5.2. Antidepressant and Antibiotic Medications

In respect to antidepressants, several studies have demonstrated that these medications may exert their protective effects, in part, by altering the richness, diversity, and composition of the GM in favor of SCFA-producing bacteria, which are known to facilitate epigenetic modifications [199,200]. In an investigation conducted by Zhang et al., the authors reported that antidepressant effects of a 6-week treatment with fluoxetine and amitriptyline in rats exposed to CUMS are linked to increased abundance of anti-inflammatory butyrate-producing bacteria including *Butyricimonas* [201]. In a human clinical study Jiang et al. reported higher levels of anti-inflammatory bacteria, including *Ruminococcus*, *Bifidobacterium*, and *Faecalibacterium*, which are involved in butyrate production, in responders to serotonin reuptake inhibitors (SSRIs) versus non-responders [202].

There is also evidence that some antibiotics with antidepressant properties my influence inflammation and the GM. For instance, Yang et al. reported that a 4-week minocycline treatment exhibited antidepressant effects in CUMS mice by suppressing neuroinflammation in the hippocampus, reshaping the GM, improving intestinal barrier integrity, and restoring butyrate concentration [203]. Li et al. demonstrated that treating rats exposed to CUMS with 150 mg/kg rifaximin (a broad-spectrum, non-absorbable antibiotic) for 4 weeks enhanced the content of anti-inflammatory mediators secreted by microglia and increased the abundance of *Ruminococcaceae* and *Lachnospiraceae*, which are positively associated with elevated butyrate concentration in the brain [204].

## 6. Conclusions and Future Perspectives

The findings described in this review indicate that disturbances in the GM composition give rise to immune imbalance and subsequently the development of NPD in part by inducing epigenetic aberrations. Knowledge about various communication pathways on the bidirectional gut–brain axis paves the way for researchers to design promising therapeutic strategies for the prevention or treatment of NPD. Specific diets, probiotic/prebiotic supplementations, and various medications are capable of modulating GM, increasing the availability of their metabolites in the intestine, improving immune system function, and hence mitigating the severity of NPD via normalizing epigenetic aberrations. Thus, further research into the interactions between the GM, the immune system, the epigenome, and the brain against a genetic backdrop paves the way for more precise understanding of the pathophysiological mechanisms of NPD and exploring novel and alternative therapeutic targets and strategies for the prevention or treatment of NPD. However, several challenges should be addressed in this field. For instance, the majority of the existing investigations have small sample, which may result in insufficient statistical power to establish robust and reproducible associations. Therefore, gaining deeper insights in the field requires future studies with larger sample sizes and more refined experimental design. To navigate the complexity of this field and develop innovative therapies, collaborative studies involving experts from diverse disciplines—including microbiology, immunology, neuroscience, and computational biology—should be pursued. To clarify the influence of microbial communities in the GI tract on immune system function in the brain, and to gain deeper understanding of the effects of GM-based therapeutics on the GM–immune system–brain axis, large longitudinal clinical studies should be conducted in individuals with NPD rather than relying on animal models.

## Figures and Tables

**Figure 1 cells-14-01027-f001:**
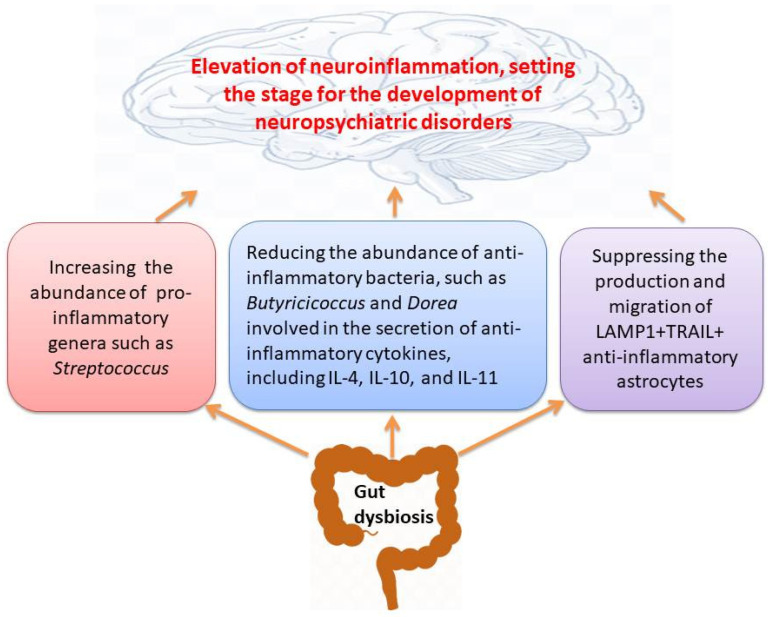
Association between changes in the gut microbiota due to gut dysbiosis and the development of neuropsychiatric disorders through increasing neuroinflammation.

**Figure 2 cells-14-01027-f002:**
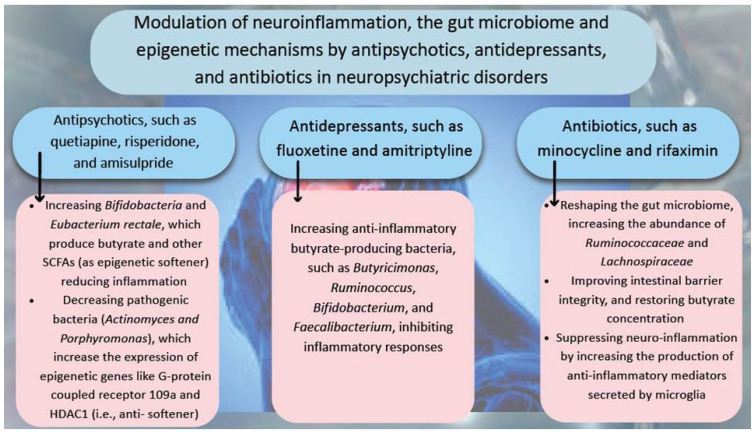
Potential mechanisms of antipsychotics, antidepressants, and antibiotics in the treatment of neuropsychiatric disorders by modulating neuroinflammation through altering the gut microbiome composition and epigenetic mechanisms.

**Table 1 cells-14-01027-t001:** Evidence supporting neuroinflammation in different neuropsychiatric disorders.

Neuropsychiatric Disorders	Study Subjects/Sample	Immune Cells	Main Outcomes	Ref.
Schizophrenia (SCZ)	Human/blood	Neutrophils and lymphocyte	Association between greater proportions of neutrophils and neutrophil-to-lymphocyte ratio and higher PANSS-total scores in SCZ	[36]
SCZ and Bipolar Disorder (BD)	Human/ postmortem midbrain tissue	Parenchymal CD163+ cells	Association between higher proportions of parenchymal CD163+ cells and CD163 protein level, and a reduction in Tyrosine Hydroxylase (TR) in the substantia nigra	[37]
Autism spectrum disorder (ASD)	Human/blood	B cells	Reduced expression of anti-inflammatory cytokine IL-10 and elevated expression of TNF-α and IL-6 due to TLR4 activation in B cells in ASD	[38]
Parkinson disease (PD)	Human/blood	Central memory CD4+ T cells, naive CD4+ and naive CD8+ T cells	Reductions in naive B cells, naive CD4+ and naive CD8+ T cells; elevation of TNF-α–producing CD19+ B cells, central memory CD4+ T cells, IL-17–producing CD4+ Th17 cells, IL-4–producing CD4+ Th2 cells, and IFN-γ–producing CD8+ T cells in PD	[39]
PD	Human/blood	CD8+ T cell and natural killer (NK) cells	Negative association between Montreal Cognitive Assessment scores and Intracellular TNF-α in naïve CD8+ T-cell cluster C16 (CD57− naïve CD8+ T) and NK cell cluster C32 (CD57− CD28− NK)	[40]
PD	Human/blood	Myeloid dendritic cells and CD27+ CD4+ memory T cells	Relation between elevated genetic risk for PD and greater levels of myeloid dendritic cells and CD27+ CD4+ memory T cells	[41]
Alzheimer’s disease (AD)	Human/blood and brain	NK cells and CD4 T cells	Reduced NK cells and increased CD4 T cells in AD	[42]
AD	Human/blood	CD4+ T, CD8+ T, B and, NK cells, monocytes–macrophages	High-frequency amplification clonotypes in T and B cells and reduced T cells diversity in AD	[43]
AD	Human/blood	Monocytes, regulatory T cells (Treg), and B cells	AD risk in linked to CD33 on CD14+ monocyte; AD risk is inversely linked to secreting CD4 regulatory T cells, %CD4 regulatory T cells and CD25 on switched memory B cells	[44]
Major depressive disorder (MDD)	Human/blood	T regulatory cells	Acute phase of severe MDD is linked to breakdown of immune tolerance, and CD40L activation; elevated levels of CD3+ CD71+, CD3+ CD40L+, CD4+ CD71+, CD4+ CD40L+, CD4+ HLADR+, and CD8+ HLADR+ T cells in MDD	[45]

**Table 2 cells-14-01027-t002:** Potential roles of inflammatory gut bacteria in the pathogenesis of specific neuropsychiatric disorders.

Neuropsychiatric Disorders	Study and Sample Type	Alerted Pro- or Anti-Inflammatory Bacteria and Their Epigenetic Metabolites	Main Finding	Ref.
Alzheimer’s disease (AD)	Human/fecal and blood samples	inflammatory bacteria such as Synergistetes and the Christensenellaceae family	Increased abundance of inflammatory bacteria and elevated levels of inflammatory cytokines in AD	[54]
AD	Human/fecal samples	Pro-inflammatory (e.g., *Escherichia*/*Shigella*, *Clostridium_sensu_stricto_1* and anti-inflammatory genera (*Faecalibacterium*, *Blautia*, *Bacteroides*, and *Roseburia*	Higher levels of pro-inflammatory bacteria and lower levels of anti-inflammatory genera and total SCFAs in AD	[55]
Parkinson’s disease (PD)	Human/ fecal samples	Anti-inflammatory butyrate-producing bacteria, including *Roseburia intestinalis*, *Faecalibacterium prausnitzii*, *Anaerostipes hadrus*, and *Eubacterium rectale*	Depletion of anti-inflammatory butyrate-producing bacteria, derangements in SCFA-synthesis, and increased neuro-inflammation due to intestinal inflammation	[56]
Autism spectrum disorder (ASD)	Human/Stool and blood samples	Anti-inflammatory bacteria like Lachnospiraceae family	Negative correlation between pro-inflammatory cytokines IFN-γ and IL-6 and Lachnospiraceae family	[57]
SCZ	Human/fecal samples	Butyrate-producing and succinate-producing bacteria (*Phascolarctobacterium succinatutens* and *Paraprevotella clara*)	Association between increased levels of succinate-producing bacteria and inflammation	[58]
SCZ	Human/fecal and blood samples	Pro-inflammatory genera such as *Proteus* and *Succinivibrio* and anti-inflammatory butyrate-producing bacteria	Reduced levels of butyrate-producing bacteria (e.g., *Faecalibacterium*, *Blautia*, *Alistipes*, *Gemmiger*, and *Butyricicoccus*) and elevated levels of genera such as *Proteus* and *Succinivibrio*; positive correlations between pro-inflammatory cytokines (IL-1β, IL-2, IL-6, and TNF-α) and *Succinivibrio*	[59]
BD	Human/stool samples	Pro-inflammatory genera, like *Streptococcus*	Association between higher IL-6 levels and greater abundance of pro-inflammatory bacteria, like *Streptococcus*	[60]
BD	Human/Stool samples	Pro-inflammatory genera, like *Flavonifractor*	Positive correlation between *Flavonifractor* and oxidative stress and inflammation	[61]
Depressive BD II	Human/fecal samples	Pro -inflammatory bacteria (e.g., *Proteobacteria*, *Enterobacteriaceae*, *Porphyromonadaceae*, and *Pseudescherichia*)	Higher levels of *Proteobacteria*, *Enterobacteriaceae*, *Porphyromonadaceae*, and *Pseudescherichia*, along with inflammatory cytokines in unmedicated depressive BD II vs. controls	[62]
Depression	Human/fecal samples	Gut anti-inflammatory (*Faecalibacterium* and *Subdoligranulum*) and pro-inflammatory (*Flavonifractor* and *Gammaproteobacteria*) *bacteria*	Decreased abundance of *Faecalibacterium* and *Subdoligranulum* and increased abundance of *Flavonifractor* and *Gammaproteobacteria* in depressed vs. control subjects	[63]
Depression	Human/fecal sample	Pro-inflammatory genera such as *Streptococcus* and anti-inflammatory genera, like *Faecalibacterium*	Elevated abundance of *Streptococcus* and *Escherichia*/*Shigella*, and reduced abundance of *Faecalibacterium*; higher levels of pro-inflammatory cytokines like IL-17, and lower levels of anti-inflammatory cytokines, like IFN-γ	[64]
Depression	Human/fecal and blood sample	Anti-inflammatory butyrate-producing bacteria such as *Turicibacter*, *Roseburia*, and *Clostridium*	Reduced levels of anti-inflammatory bacteria; negative correlation between *Turicibacter* and *Turicibacteraceae* and IL-1β and IL-6 levels	[65]
Inflammatory depression	Human and mouse/fecal, blood, and colon biopsy samples	Anti-inflammatory bacteria such as *Clostridium* and *Faecalibacterium*	Elevated levels of *Bacteroides* and reduced levels of Clostridium and Faecalibacterium in inflammatory vs. non-inflammatory depression and HCs; lower levels of propionic and butyric acids in depressed patients vs. HCs	[66]
Depression	Human/fecal samples	Pro-inflammatory genera such as *Streptococcus* and anti-inflammatory genera, like *Faecalibacterium*	Lower α-diversity and richness, changes in β-diversity, elevated abundance of *Streptococcus* and reduced abundance of *Faecalibacterium*	[67]

**Table 3 cells-14-01027-t003:** Associations between altered DNA methylation patterns in immune-relevant genes and developing specific neuropsychiatric disorders in humans.

Neuropsychiatric Disorders	Type of Sample/Study Population	Immune-Relevant Genes	Key Findings	Ref.
Acute mania	Serum samples/20 mania and 20 unaffected controls	CYP11A1	Relationship between methylation of CYP11A1 and three inflammatory markers in patients	[75]
Schizophrenia (SCZ)	Blood/deficit SCZ (n = 53), non-deficit SCZ (n = 55), and 63 healthy controls (HCs)	CXCL1	Hypomethylation of most CpG sites within CXCL1 gene in SCZ vs. HCs	[76]
Psychosis	Leukocyte/60 non-affective psychosis and 40 HCs	DDR1	Association between DDR1 hypermethylation and inflammatory markers	[77]
SCZ	Peripheral blood cells (PBCs)/monozygotic twins discordant for SCZ	SOCS3 and CASP1	Reactivating a SOCS3-mediated anti-inflammatory response by LncRNA-AC006129.1 via DNA methylation-mediated down-regulation of Capicua gene	[78]
SCZ	PBCs/469 Han Chinese patients with SCZ	The Th1 regulatory-related genes (SLC11A1, TNFSF4, IL27, and IL1R1); L12B, IL27, S100A12, and ZAP70	Symptom severity is linked to DNA methylation of immune-relevant genes; hypermethylation of L12B, IL27, S100A12, and ZAP70 correlate to better response to antipsychotics	[79]
Bipolar disorder (BD)	PBCs/84 BD subjects with a history of suicide attempt (SA) (BD + SA), 79 BD subjects without history of SA (BD–SA)	CXCL8, CD300LG, LFNG, TRIM40, RNF14, and HIVEP3	Six differentially methylated positions (DMPs) and seven differentially methylated regions (DMRs) in BD + SA vs. BD–SA in immune-related genes	[80]
BD	Leukocyte/128 BD patients in remission and 141 HCs	DDR1	DDR1 hypermethylation at cg19215110 and cg23953820, and hypomethylation at cg14279856 and cg03270204 sites are linked to immune and inflammatory mechanisms in BD	[81]
Major depressive disorder (MDD)	Whole blood and serum samples/self-reported history of depression (n = 100) vs. no depression (n = 100)	LTB4R2 and IL-6	Six DMRs in exon 1 of LTB4R2 gene; one depression-associated co-methylation module relevant to telomere length and IL-6 levels	[82]
MDD	Blood/153 subjects with MDD	IL1-β and IL6R	Higher methylation percentage of treatment responders in an IL6R CpG island	[83]
MDD	Blood/52 young patients with MDD in Scandinavian adults	TLR4	Reduced methylation of TLR4 in blood is linked to greater depression scores	[84]
MDD	220 MDD and 82 HCs	NLRP3	DMPs in NLRP3 are linked to brain structural alterations (NLRP3 DNA methylation may elevate NLRP3 inflammasome-related neuroinflammation)	[85]
Alzheimer’s disease (AD)	Human brain/5 from AD and age-matched non-dementia controls	CASPASE-4 (CASP4)	Hypomethylation of CASP4 in AD, and elevated expression of CASP4, and IL-1β	[86]
Autism spectrum disorder (ASD)	Peripheral blood neutrophils/52 ASD children and 24 controls	CCR2 and MCP-1	DNA hypo-methylation and increased levels of inflammatory mediators (CCR2 and MCP-1)	[87]

## Data Availability

Not applicable.

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
