# Peer review of "Therapeutic Horizons: Gut Microbiome, Neuroinflammation, and Epigenetics in Neuropsychiatric Disorders"

_cells, 2025, doi:10.3390/cells14131027_

Round 1
Reviewer 1 Report
Comments and Suggestions for Authors
The review article discusses the relationship among the GM, neuroinflammation and neuropsychiatric diseases through the GM-immune-brain axis. It is well-written and easy to follow. References are used effectively. Tables and figures summarize the main points appropriately. The topic is attractive to the readers in the field. No major issues have been identified and the manuscript is ready for publication.
Author Response
Dear Ms. Kathy
Section Managing Editor of cells,
Thank you for providing the reviews for our manuscript and sending us the reports. We greatly appreciate the valuable feedback from the reviewers and have carefully considered all their comments. We have revised our manuscript, accordingly, addressing each point raised. There are no comments for reviewer 1 since the reviewer recommended acceptance in present form.
In the revised version, amendments made in response to the second reviewer's comments are highlighted in green and amendments made in response to third Reviewer's comments are highlighted in blue.
We believe the reviewers feedbacks and recommendations included in the revised manuscript have significantly improved the quality and clarity of this work. We hope that our revised version will now be considered suitable for publication in your esteemed journal. Please find below a point-by-point response to the reviewers' comments.
Best regards,
Dr. Shabnam Nohesara
Reviewer 1
Open Review
(x) I would not like to sign my review report
( ) I would like to sign my review report
Quality of English Language
( ) The English could be improved to more clearly express the research.
(x) The English is fine and does not require any improvement.
Is the work a significant contribution to the field?
Is the work well organized and comprehensively described?
Is the work scientifically sound and not misleading?
Are there appropriate and adequate references to related and previous work?
Comments and Suggestions for Authors
The review article discusses the relationship among the GM, neuroinflammation and neuropsychiatric diseases through the GM-immune-brain axis. It is well-written and easy to follow. References are used effectively. Tables and figures summarize the main points appropriately. The topic is attractive to the readers in the field. No major issues have been identified and the manuscript is ready for publication.
We sincerely appreciate your valuable comments on our manuscript. We also revised the manuscript based on the valuable comments of reviewers 2 and 3
Reviewer 2 Report
Comments and Suggestions for Authors
The authors reviewed the most recent articles related to relation of major mental illnesses with neuroinflammation, gut microbiome, and epigenetics. The article is well written and organized. I have only a few minor comments for consideration:
- Throughout the manuscript, the authors presented findings related to changes in inflammatory markers and gut bacteria in different mental illnesses. However, they did not comment which alternations in inflammation and gut microbiota are specifically related to the particular condition, and which are shared across different mental conditions. Please, comment on this subject since this is of particular importance when considering therapeutic strategies for curing different disorders.
- Lines 519-521: In the section Antidepressant and antibiotic medications, the authors stated: “In respect to antidepressants, several studies have demonstrated that these drugs may exert their protective effects in part by altering the richness, diversity, and composition of inflammatory gut bacteria in the gut via epigenetic mechanisms [190,191]”. However, these articles did not examine epigenetic mechanisms. Instead, they demonstrated that the therapeutic effects of antidepressants are mediated through changes in gut microbiota, including proinflammatory and SCFA-producing bacteria. Please, revise this statement.
- In the section Prebiotics/ postbiotics, the authors did not specify that SCFAs are postbiotics. Please, clarify this.
- Minor grammatical corrections:
- Lines 314, 259, 369: please, correct “depressive like” to “depressive-like“.
- Line 372: please, define abbreviation “PFC”.
- Line 449: there is no need to define abbreviation for KD again.
Author Response
Reviewer 2
Open Review
(x) I would not like to sign my review report
( ) I would like to sign my review report
Quality of English Language
( ) The English could be improved to more clearly express the research.
(x) The English is fine and does not require any improvement.
Is the work a significant contribution to the field?
Is the work well organized and comprehensively described?
Is the work scientifically sound and not misleading?
Are there appropriate and adequate references to related and previous work?
Comments and Suggestions for Authors
The authors reviewed the most recent articles related to relation of major mental illnesses with neuroinflammation, gut microbiome, and epigenetics. The article is well written and organized. I have only a few minor comments for consideration:
We sincerely appreciate your valuable comments. We have carefully revised the manuscript in accordance with your suggestions. Appropriate amendments have been made, and additional information has been included, all of which are highlighted in green for your convenience.
Throughout the manuscript, the authors presented findings related to changes in inflammatory markers and gut bacteria in different mental illnesses. However, they did not comment which alternations in inflammation and gut microbiota are specifically related to the particular condition, and which are shared across different mental conditions. Please, comment on this subject since this is of particular importance when considering therapeutic strategies for curing different disorders.
Author response: in the introduction and section 2 as well as figure 1, we described general and shared feature of neuropsychiatric disorders in respect to inflammation, epigenetic alterations and gut dysbiosis. In tables 1-3 we presented disease specific changes, related to inflammatory, epigenetic and microbiome alteration. We modified text (please see figure 1 and Table 1, 2 and 3) to address general and disease specific alterations.
Lines 519-521: In the section Antidepressant and antibiotic medications, the authors stated: “In respect to antidepressants, several studies have demonstrated that these drugs may exert their protective effects in part by altering the richness, diversity, and composition of inflammatory gut bacteria in the gut via epigenetic mechanisms [190,191]”. However, these articles did not examine epigenetic mechanisms. Instead, they demonstrated that the therapeutic effects of antidepressants are mediated through changes in gut microbiota, including proinflammatory and SCFA-producing bacteria. Please, revise this statement.
Author response: we revised the section to “Antidepressant and antibiotic medications to “In respect to antidepressants, several studies have demonstrated that these medications may exert their protective effects, in part, by influencing the richness, diversity, and composition of the gut microbiota in favor of SCFA-producing bacteria, which are known to facilitate epigenetic modifications”.
Please see page 22 lines 537-541 highlighted in green.
In the section Prebiotics/ postbiotics, the authors did not specify that SCFAs are postbiotics. Please, clarify this.
Author response: Relevant information has been added to the manuscript in response to this comment, highlighted in green, page 18, lines 365-367.
Minor grammatical corrections:
Lines 314, 259, 369: please, correct “depressive like” to “depressive-like“.
Author response and action taken: We implemented the comment in the revised version
Line 372: please, define abbreviation “PFC”.
Author response and action taken: We implemented the comment in the revised version
Line 449: there is no need to define abbreviation for KD again.
Author response and action taken: We implemented the comment in the revised version
Reviewer 3 Report
Comments and Suggestions for Authors
The review "Therapeutic Horizons: Gut Microbiome, Neuroinflammation, and Epigenetics in Mental Illness,” by Nohesara et al., submitted to Cells, explores the interaction between gut health, epigenetics, and inflammation and how compromises in these systems may contribute to the development of neuropsychiatric disorders. The author covers all the points presented in the abstract. The review is organized and discusses the most recent findings in the field; however, there are concerns.
Title:
- Alzheimer's disease and Parkinson's disease are not considered mental illnesses but neurodegenerative diseases. The authors should change the title to reflect that they are discussing psychiatric disorders and neurogenerative diseases. The term neuropsychiatric disorders will reflect both instead of the term mental illness.
- This title change will, of course, lead to mental illness, not referring to all neuropsychiatric disorders discussed throughout the review. Instead of the authors using mental illness (MI) throughout the review, the authors should change it to neuropsychiatric disorders throughout the review.
Introduction:
- The authors should briefly discuss that neuropsychiatric disorders consist of mental disorders (e.g., anxiety, depression, ADHD) and neurogenerative diseases (e.g., Alzheimer’s, Parkinson's), and how mental disorders and neurogenerative diseases may differ from each other.
Main body:
- All abbreviations should be defined before being used.
- Double-check spelling and grammar.
Tables:
- Table 1 in the headings change “mental disorders” to neuropsychiatric disorders.
- In Table 2, change the heading from “disease type” to “neuropsychiatric disorders”.
- Table 3 in the headings change “mental illness” to neuropsychiatric disorders.
Author Response
Reviewer 3
Open Review
(x) I would not like to sign my review report
( ) I would like to sign my review report
Quality of English Language
( ) The English could be improved to more clearly express the research.
(x) The English is fine and does not require any improvement.
Is the work a significant contribution to the field?
Is the work well organized and comprehensively described?
Is the work scientifically sound and not misleading?
Are there appropriate and adequate references to related and previous work?
Comments and Suggestions for Authors
The review "Therapeutic Horizons: Gut Microbiome, Neuroinflammation, and Epigenetics in Mental Illness,” by Nohesara et al., submitted to Cells, explores the interaction between gut health, epigenetics, and inflammation and how compromises in these systems may contribute to the development of neuropsychiatric disorders. The author covers all the points presented in the abstract. The review is organized and discusses the most recent findings in the field; however, there are concerns.
We highly appreciate your feedback on our manuscript. We have revised the manuscript based on your valuable comments. Please see blue highlights.
Title:
Alzheimer's disease and Parkinson's disease are not considered mental illnesses but neurodegenerative diseases. The authors should change the title to reflect that they are discussing psychiatric disorders and neurogenerative diseases. The term neuropsychiatric disorders will reflect both instead of the term mental illness.
This title change will, of course, lead to mental illness, not referring to all neuropsychiatric disorders discussed throughout the review. Instead of the authors using mental illness (MI) throughout the review, the authors should change it to neuropsychiatric disorders throughout the review.
Author response: we revised the title and changed it to neuropsychiatric disorders throughout the review
Introduction:
The authors should briefly discuss that neuropsychiatric disorders consist of mental disorders (e.g., anxiety, depression, ADHD) and neurogenerative diseases (e.g., Alzheimer’s, Parkinson's), and how mental disorders and neurogenerative diseases may differ from each other.
Author response: Relevant information has been added to the manuscript in response to this comment, highlighted in blue under introduction, page 1- 2, lines 38-51.
Main body:
All abbreviations should be defined before being used.
Author response and action taken: We checked it again.
Double-check spelling and grammar.
Author response and action taken: Thanks for this constructive comment. Based on your comment, the English language of the whole manuscript has been meticulously revised (grammatically, punctuation, spelling mistakes and rephrasing of the confusing sentences) and thus the quality of the manuscript has been improved.
Tables:
Table 1 in the headings change “mental disorders” to neuropsychiatric disorders.
Author response and action taken: We revised it. Please check Table 1
In Table 2, change the heading from “disease type” to “neuropsychiatric disorders”.
Author response and action taken: We implemented the comment in the revised version please check Table 2
Table 3 in the headings change “mental illness” to neuropsychiatric disorders.
Author response and action taken: We implemented the comment in the revised version please check Table 3
Round 2
Reviewer 3 Report
Comments and Suggestions for Authors
All the concerns have been addressed.